

# Design water demand of irrigation for a large region using a high-dimensional Gaussian copula

Xinjun Tu[1,2], Yiliang Du[1], Vijay P Singh[3,4], Xiaohong Chen[1,2], Kairong Lin[1], Haiou Wu[1]

[1]Department of Water Resources and Environment, Sun Yat-sen University, Guangzhou, 510275, China
[2]Center of Water Security Engineering and Technology in Southern China of Guangdong, Guangzhou, 510275, China
[3]Department of Biological and Agricultural Engineering, Texas A&M University, 2117 College Station, Texas, 77843, USA
[4]Zachry Department of Civil Engineering, Texas A&M University, 2117 College Station, Texas, 77843, USA

*Correspondence to*: Xinjun Tu (eestxj@mail.sysu.edu.cn)

**Abstract.** Spatial and frequency distributions of precipitation should be considered in determining design water demand of
irrigation for a large region. In Guangdong province, South China, as a study case, an eight-dimensional joint distribution of
precipitation for agricultural sub-regions was developed. A design procedure for water demand of irrigation for a given
frequency of precipitation of the entire region was proposed. Water demands of irrigation in the entire region and its sub-
regions using three design methods, i.e. equalized frequency (EF), typical year (TY) and most-likely weight function (MLW),
were compared. Results demonstrated that the Gaussian copula efficiently fitted the high-dimensional joint distribution of
eight sub-regional precipitation values. The Kendall frequency was better than the conventional joint frequency to analyze the
linkage between the frequency of the entire region and the joint frequency of sub-regions. For given frequencies of precipitation
of the entire region, design water demands of irrigation of the entire region among the MLW, EF and TY methods slightly
differed, but those of individual sub-regions of the MLW and TY methods fluctuated around the demand lines of the EF
method. The alterations of design water demand in sub-regions were more complicated than those in the entire region. The
design procedure using the MLW method in association with a high-dimensional copula, which simulated individual univariate
distributions, captured their dependences for multi-variables, and built a linkage between regional frequency and sub-regional
frequency of precipitation, is recommended for design water demand of irrigation for a large region.

## 1 Introduction

Water demand of irrigation in a region is associated with exploitation and utilization of land and water resources, such as farm
area, cultivation pattern, category of crops, canal technology etc., and is also impacted by natural factors, for example
precipitation volume and soil properties (Tarjuelo et al., 2005; Griffin, 2006; Leenhardt et al., 2004, 2011; Wriedt et al., 2009;
Davidson and Hellegers, 2011). Among the many factors, precipitation which is regarded as stochastic is an uncertain factor
influencing irrigation (Wisser et al., 2008; Thomas, 2008; Cai et al., 2011; Meza et al., 2012). In general, the more precipitation
is, the less water demand of irrigation is, and vice versa. In practical regional water resources planning, water demands of
irrigation are estimated in association with various frequencies of regional precipitation (Gohari et al., 2013).



For design water demand in a large region, it is important that the heterogeneity of those factors in geographical space is considered (Lankford, 2010; Leenhardt et al., 2011). Hence, water demand should generally be analyzed in separate sub-regions in which the factors are homogenous, and can eventually be summed up. However, for a given precipitation frequency of the entire region in the design water demand of irrigation, the difficulty is how to obtain a reasonable combination of

precipitation frequencies of multiple sub-regions, even though other factors influencing irrigation have previously been demonstrated in water resources planning.

A method, named typical year (TY), has been used for design water demand of irrigation in China (Cai et al., 2001). A combination of observed sub-regional precipitation in a certain year in which the precipitation frequency for the entire region weighted by individual sub-regions was the nearest to the given frequency, was selected and zoomed in or out. Due to the

limited observed samples, the representation of the typical year has been questionable. In actuality, the frequency of regional precipitation in a large region corresponds to various combinations of sub-regional frequencies.

Moreover, design water demand of irrigation in a large region not only presents the stochastic characteristic of precipitation in the entire region and its sub-regions, but also addresses the relationship between sub-regional precipitation as well as heterogeneity. The multi-variable statistical simulation and joint design theory can be considered to improve the selection of

an appropriate combination. Due to the multivariate dependence and the flexibility of unbounded marginal distributions, copula functions have been widely applied in the simulation and design of hydrological multi-variables, e.g. extreme properties of heavy precipitation (Zhang et al., 2011, 2013; Abdul Rauf and Zeephongsekul, 2014), floods (Zhang and Singh, 2006; Chowdhary et al., 2011; Zhang et al., 2015a) and droughts (Ganguli and Reddy, 2014; Zhang et al., 2015b; Tu et al., 2016). In recent years, they have been used to analyze floods or droughts encountered in multiple hydrological regions in China (Yan et

al., 2010; Xie et al., 2012; Tu et al., 2017). These studies on multivariate hydrology mostly focused on bivariate and trivariate issues, but less on higher dimensional hydrological statistical analyses (Liu and Menzel, 2016; Chen et al., 2015). A high-dimensional meta-Gaussian copula beyond three variables has been applied in other fields, e.g. economic analysis. (Aussenegg and Cech, 2012; Creal and Tsay, 2015).

As a multidimensional copula and its marginal distributions are determined according to observed samples, the joint frequency

and probability density for any combination of precipitation frequencies of sub-regions can be calculated. In a practical design combination, a large quantity of combinations can be randomly generated using the Monte Carlo method on the basis of the determined copula. It is clear that a given joint frequency corresponds to quite a number of combinations. In order to select one from various combinations, a simple method, based on the equalized frequency (EF) method, which refers to all marginal frequencies being identical, is used for design flood peak and volume (Liu et al., 2015).

Another improved method is that the combination can be exclusively determined by using the most-likely weight function (MLW) in association with the products of joint and marginal probability densities (Salvadori et al., 2011). The most-likely weight design method has been applied for the design combination of hydrological multivariables, such as flood and drought properties (van den Berg et al., 2012; Zhang et al., 2015c), flood and tide or heavy precipitation and tide in coastal rivers





(Corbella and Stretch, 2012; Lian et al., 2013; Zheng et al., 2013), as well as precipitation or streamflow of multiple regions (Yan et al., 2010).

Furthermore, in practical water resources planning, the main concern is that water demand of irrigation of the entire region and that of sub-regions are designed for a given frequency of precipitation for the entire region, not for a given joint frequency of precipitation of sub-regions. The relationship between regional frequency and joint frequency of precipitation is also required to be investigated in determining design water demand of irrigation in a large region.

This paper considered water demand of irrigation of paddy in Guangdong province, South China, as a case study of a large region. Annual precipitation data of eight agricultural sub-regions and their net quotas of irrigation in association with precipitation frequency were used. A high-dimensional meta-Gaussian copula and several conventional univariate distributions were applied to fit the joint and marginal distributions of sub-regional precipitation, respectively. Combinations of cumulative frequency for sub-regions precipitation were generated by using the Monte Carlo simulation method on the basis of the determined copula. The link between the joint frequency of sub-regional precipitation and the frequency of precipitation of the entire region was established. The methods, i.e. typical year, equalized frequency, and the most-likely weight function, were used for design combinations of sub-regional precipitation for given frequencies of precipitation of the entire region. Water demand of irrigation of the entire region and individual sub-regions among design methods were compared in order to improve design water demand of irrigation in a large region.

## 2 Methodology

### 2.1 Water demand of irrigation of paddy

In a paddy field, water demand of irrigation per unit area, i.e. the net quota of irrigation, is determined by various factors, such as crop and soil types, cultivation pattern and precipitation process etc. In regional planning of water resources, as other factors are demonstrated in advance, the net quota of irrigation mainly changes with annual precipitation due to the stochastic property of precipitation. In practice, the net quota of irrigation per unit area $q(u)$ in association with the frequency of precipitation is determined via field experiments. Then, for a given frequency of annual precipitation, $u$, water demand of irrigation, $W(u)$, which refers to water withdrawn from river or other water sources engineering, can be calculated as:

$$W(u) = \frac{Aq(u)}{\eta},\tag{1}$$

where $A$ and $q(u)$ refer to the paddy area and the net quota of irrigation, respectively, and $\eta$ is the utilization coefficient of irrigation which refers to the ratio of net water supply in the field to water withdrawn from river or other water sources engineering.



In a large region, for example Guangdong province, China, the differences of annual precipitation and the net quota of irrigation among individual agricultural regions are required to be considered. Then, for a given regional frequency of annual precipitation, $u_0$, for a large region with   sub-regions, water demand of irrigation, $W(u_0)$, can be deduced as follows:

$$W(u_0) = \sum_{i=1}^{d} \frac{A_i q_i(u_i)}{\eta},$$ (2)

where $A_i$ refers to the irrigation area of the $i$ th sub-region and $q_i(u_i)$ is the net quota of irrigation per unit area at the frequency of precipitation of the $i$ th sub-region, $u_i$.

Therefore, the key point to design water demand of irrigation for a given precipitation frequency of the entire region, $u_0$, is how to determine a combination of sub-regional precipitation frequencies, $\{u_1, \cdots, u_d\}$.

## 2.2 Annual precipitation distribution

If a large region consists of $d$ sub-regions, let $X_i$ be the series of annual precipitation in which $i = 1, \cdots, d$ and $j = 1, \cdots, n$ refer to sub-region $d$ and $n$ years, respectively. The theoretical cumulative distribution of annual precipitation in a sub-region, $F_{X_i}(x)$ can be fitted by several conventional three parameter univariate distributions, which have been widely used in hydrology, such as generalized extreme value (GEV), generalized logistic (GLO), Pearson III (P-III), and generalized normal (GNO). Their cumulative distribution and probability density functions are presented in Table 1.

The Kolmogorov-Smirnov (K-S) statistic, $D$, for the goodness-of-fit test of annual precipitation distribution of a sub-region was computed as (Dobric and Schmid 2007; Massey, 2012; Tu et al. 2016):

$$D = \max_{1 \leq j \leq n} \left\{ F[x(j)] - \frac{j-1}{n}, \frac{j-1}{n} - F[x(j)] \right\},$$ (3)

The parameters of all recommended univariate distributions were estimated by the L-moment method. The critical values at the significance level of 0.05 for the goodness-of-fit test of all distributions were obtained by the Monte Carlo method with 20   5000 or more simulations. If the K-S statistic, $D$, which was computed from the samples, was less than the corresponding critical value, the tested distribution was accepted. The optimal distribution was selected from the accepted distributions by comparing their root-mean-square error (RMSE) and Akaike information criterion (AIC) values. In addition, empirical and theoretical distributions were compared to evaluate the goodness of fit to the observed samples of precipitation. In hydrological practice, the empirical distribution functions are defined and transformed by Gringorten (1963).

Subsequently, based on the areal weight method, the annual precipitation in the entire region, $X_0(j)$ was calculated as:

$$X_0(j) = \sum_{i=1}^{d} \alpha_i X_i(j),$$ (4)

Where $\alpha_i$ refers to the areal ratio of the $i$ -th sub-region to the entire region. Then, the theoretical distribution of annual precipitation for the entire region, $F_{X_0}(x)$, can also be fitted by using the above recommended univariate distributions and the K-S test method.





## 2.3 Conventional design methods

### 2.3.1 Typical year method (TY)

In practical design, water demand of irrigation for a large region, a combination of observed sub-regional precipitation in a certain year, in which the precipitation of the entire region weighted by individual sub-regions is the nearest to that of a given

frequency for the entire region, has been the only selection (Cai et al., 2001). The selected year was called the typical year corresponding to the given frequency of precipitation. Let $\tilde{u}_0$ be a given cumulative distribution frequency (CDF) of precipitation of the entire region. Then, the corresponding precipitation can be calculated by using the inverse function of the frequency distribution as follows:

$$\tilde{X}_0 = F_{X_0}^{-1}(\tilde{u}_0) \,, \tag{5}$$

Further, the relative alteration, $R(j), j = 1, \cdots, n$ of the observed precipitation in each year compared to the precipitation of the given frequency, was defined as:

$$R(j) = \left| \frac{X_0(j) - \tilde{X}_0}{\tilde{X}_0} \right|, \tag{6}$$

Then,

$$J = \operatorname{argmin} R(j) \,, \tag{7}$$

where $J$ is the selected rank of a certain year, which corresponds to the typical year for the given frequency $\tilde{u}_0$.

In addition, due to the limited length of annual precipitation observations, the relative alteration $R(J)$ of the typical selected year might be large. Namely, annual precipitation of the entire region weighted by sub-regional precipitation in a typical selected year, in terms of magnitude, may differ from that of the given frequency. Therefore, a scaling method was applied to zoom in or out for the sub-regional precipitation in the typical selected year as follows:

$$\beta = \frac{X_0(J)}{\tilde{X}_0}, \tag{8}$$

$$\tilde{X}_i(J) = \frac{X_i(J)}{\beta}, \tag{9}$$

where $\beta$ is a scale coefficient, $X_i(J)$ and $\tilde{X}_i(J)$ refer to the $i$ th sub-regional precipitation before and after zooming in or out, respectively.

For a given precipitation frequency of the entire region, $\tilde{u}_0$, individual sub-regional frequencies, $\tilde{u}_i(J), i = 1, \cdots, d$ , can

eventually be deduced by their frequency distributions as follows:

$$\tilde{u}_i(J) = F_{X_i}[X_i(J)], \tag{10}$$



### 2.3.2 Equalized frequency method (EF)

In order to get a combination of sub-regional frequencies for a given precipitation frequency of the entire region, the equalized frequency method (Liu et al., 2015), is also used for downscaling precipitation for a large region. As the name implies, the EF assumes that the frequencies of sub-regional precipitation are identical. That is, for a given precipitation frequency $\tilde{u}_0$, let
$\tilde{u}_1 =, \cdots, = \tilde{u}_d$, and then $\tilde{u}_i$ can be found as follows:

$$F_{X_0}^{-1}(\tilde{u}_0) = \sum_{i=1}^{d} A_i F_{X_i}^{-1}(\tilde{u}_i) , \tag{11}$$

where $F_{X_0}^{-1}(\tilde{u}_0)$ and $F_{X_i}^{-1}(\tilde{u}_i)$ refer to the precipitation in the entire region and sub-regions calculated by using the inverse function of individual frequency distributions, respectively. In practical design, the sub-regional frequency $\tilde{u}_i$ is determined by applying the method of successive search approximation within an available range.

## 2.4 Joint design based on copula function

### 2.4.1 Multidimensional joint distribution

For a large region, the difference of precipitation between any two sub-regions is associated with geographical location. Though annual precipitation in any sub-region is regarded as stochastic, there exists dependence between two sub-regions, in particular for adjacent sub-regions due to similar geographic and climate conditions. Herein, a multidimensional copula
function was used for modeling the joint distribution of sub-regional precipitation. Assume that annual precipitation of each sub-region, $X_i, i = 1, \cdots, d$, is continuous random variable with a $d$-dimensional joint distribution $H(X_i, \cdots, X_d)$ and individual marginal distribution functions $F_{X_i}(x), i = 1, \cdots, d$. Then, on the basis of the Sklar theorem (Nelsen, 2006), the joint distribution, $H(X_1, \cdots, X_d)$, can be defined as:

$$H(X_1, \cdots, X_d) = C[F_{X_1}(x), \cdots, F_{X_d}(x)] = C(u_1, \cdots, u_d), \tag{12}$$

where $C(u_1, \cdots, u_d)$ is the $d$-dimensional copula function which is the joint distribution function of standard uniform random variables, and $u_i = F_{X_i}(x), i = 1, \cdots, d$, refer to individual CDFs of sub-regional precipitation.

The copula function, which accommodates different marginal distributions of individual variables and captures their dependence, has been widely applied in multivariate hydrology. More details on the theoretical properties of various copula families can be found in Nelsen (2006). Owing to its flexibility, accessibility and simple copula parameters in association with
a correlation coefficient matrix, a $d$-dimensional meta-Gaussian copula was selected for modeling the joint distribution of multiple sub-regional precipitations. Its theoretical cumulative distribution function, $C(u_1, \cdots, u_d)$, and density function, $c(u_1, \cdots, u_d)$, were deduced as follows (Genest et al., 2007):

$$C(u_1, \cdots, u_d) = \int_{-\infty}^{b_1} \cdots \int_{-\infty}^{b_d} g(\omega_1, \cdots, \omega_d) d\omega_1, \cdots, d\omega_d, \tag{13}$$




$$c(u_1, \cdots, u_d) = |\Sigma|^{-1/2} \exp\left(-\frac{\zeta^T \Sigma^{-1} \zeta}{2} + \frac{\zeta^T \zeta}{2}\right), \tag{14}$$

where

$$g(\omega_1, \cdots, \omega_d) = (2\pi)^{-d/2} |\Sigma|^{-1/2} \exp\left(-\frac{\omega^T \Sigma^{-1} \omega}{2}\right), \tag{15}$$

where $b_1 = \Phi^{-1}(u_1), \cdots, b_d = \Phi^{-1}(u_d)$, in which $\Phi^{-1}(\cdot)$ refers to the inverse function of the standard normal distribution.

$\omega = [\omega_1, \cdots, \omega_d]^T$ and $\zeta = [b_1, \cdots, b_d]^T$ are the matrices of variables in the integrand. The correlation coefficient matrix $\Sigma$ was expressed as:

$$\Sigma = \begin{bmatrix} 1 & \cdots & \rho_{1d} \\ \vdots & \ddots & \vdots \\ \rho_{d1} & \cdots & 1 \end{bmatrix}, \rho_{ij} = \begin{cases} 1, i = j \\ \rho_{ji}, i \neq j \end{cases}, \tag{16}$$

where, $\rho_{ij} \in [-1,1]$ refers to the correlation coefficient between any two sub-regional precipitations.

For the goodness-of-fit test of multi-dimensional meta-Gaussian copula, the Cramér-von Mises test statistic on the basis of the

Rosenblatt transform was used (Rosenblatt, 1952; Genest et al., 2009). For the joint distribution of sub-regional precipitation with a $d$-dimension, the goodness-of-fit test statistics, $S_n^B$, was formulated as:

$$S_n^B = \frac{n}{3^d} - \frac{1}{2^{d-1}} \sum_{j=1}^{n} \prod_{i=1}^{d} [1 - E_i^2(j)] + \frac{1}{n} \sum_{k=1}^{n} \sum_{j=1}^{n} \prod_{i=1}^{d} \{1 - \max[E_i(j), E_i(k)]\}, \tag{17}$$

where $E_i, i = 1, \cdots, d$, refers to the pseudo-observations of individual sub-regional precipitation. Let $E_i = u_1$, and $E_i, i = 2, \cdots, d$ be assigned as (Rosenblatt, 1952; Dobric and Schmid, 2007):

$$E_i = C(u_i | u_1, \cdots, u_{i-1}) = \frac{\partial^{i-1} C(u_1, \cdots, u_i)}{\partial u_1 \cdots \partial u_{i-1}} \bigg/ \frac{\partial^{i-1} C(u_1, \cdots, u_{i-1})}{\partial u_1 \cdots \partial u_{i-1}}, \tag{18}$$

A parametric bootstrap procedure for $S_n^B$, deduced from the literature, is addressed in Appendix D (Genest et al., 2009).

In addition, the Kendall function, which is a univariate expression of multivariate information (Genest and Rivest, 1993; Barbe et al., 1996; Salvadori et al., 2011, 2013), has been shown to be an appropriate tool for calculating the copula-based joint frequency of multivariate events (Nappo and Spizzichino, 2009; Salvadori et al. 2011) and is widely applied in discussing the

joint probability or return period of hydrological multivariables (Salvadori et al., 2004; De Michele et al., 2013). The Kendall CDF, $F_{K_c}$, which was transformed from the joint CDF of eight sub-regional precipitation and was used in comparing with the frequency of entire regional precipitation, was estimated as:

$$F_{K_c}(q) = P[C[u_1(j), \cdots, u_d(j)] \leq q] = \frac{1}{n} \sum_{j=1}^{n} I(C \leq q), \tag{19}$$

where $q \in (0,1)$ is the probability level, and $n$ refers to the length of observed or simulated samples. The function $I(\cdot)$ is an

indicator function, which is equal to one when the enclosed expression is true, and zero otherwise.





### 2.4.2 Most-likely weight function (MLW)

In multivariate design practice, using sample data of annual precipitation of all sub-regions, the univariate distribution of the entire region, joint and marginal distributions of sub-regions, and parameters of all distributions can be determined on the basis of the previously mentioned modelling methods. The Monte Carlo method can be used to simulate new combinations of CDFs of precipitation using the determined distributions and parameters. Then, the CDFs of precipitation of the entire region corresponding to each combination of sub-regional CDFs can be achieved. However, there are a large number of combinations which lead to the CDFs of the entire region which can be almost identical within a predefined small difference. That is, a given CDF of precipitation of the entire region can correspond to many combinations of sub-regional CDFs with enough more simulations. The design realization using the most-likely weight function was proposed by Salvadori et al. (2011) as follows:

$$[\tilde{u}_1, \cdots, \tilde{u}_d] = \mathrm{argmax} f(x_1, \cdots, x_d) \tag{20}$$

$$f(x_1, \cdots, x_d) = c(u_1, \cdots, u_d) f(x_1) \cdots f(x_d) \tag{21}$$

where $[\tilde{u}_1, \cdots, \tilde{u}_d]$ is eventually selected as the design combination of CDFs of sub-regional precipitation for a given CDF of the entire region, $\tilde{u}_0$. $f(x_1, \cdots, x_d)$ refers to the product of joint probability densities, $c(u_1, \cdots, u_d)$, and their individual marginal probability densities, $f(x_i), i = 1, \cdots, d$. Therefore, the procedure for design combination of sub-regional precipitation for a given CDF of the entire region was as follows:

**(1)** The joint and marginal distributions and parameters of sub-regional precipitation were determined by the goodness-of-fit of the recommended $d$-dimensional meta-Gaussian copula and univariate distributions for the observed samples.

**(2)** Using the Monte Carlo method according to the determined $d$-dimensional joint distribution, large quantities of combinations of CDFs of sub-regional precipitation, $[u_1(j), \cdots, u_d(j)], j = 1, \cdots, m$, with the number of simulations, $m$, were generated, and the corresponding combinations of sub-regional precipitation, $[X_1(j), \cdots, X_d(j)], j = 1, \cdots, m$, and precipitation and CDFs of the entire region, $X_0(j), j = 1, \cdots, m$ and $u_0(j), j = 1, \cdots, m$, were calculated.

**(3)** For a given CDF of precipitation of the entire region, $\tilde{u}_0$, the allowable relative error, was defined as:

$$\left| \frac{u_0(j) - \tilde{u}_0}{\tilde{u}_0} \right| \le R_e, \tag{22}$$

where $R_e$ refers to the threshold value of allowable relative error. Then, in the simulated $u_0(j)$, those which satisfied Eq. (22) were selected. That is, $u_0(k), k = 1, \cdots, l$ with the length of $l$ were found to satisfy $u_0(k) \cong \tilde{u}_0$ and the selected combinations, $[u_1(k), \cdots, u_d(k)], k = 1, \cdots, l$ and $[X_1(k), \cdots, X_d(k)], k = 1, \cdots, l$ corresponded to the given $\tilde{u}_0$.

**(4)** For all selected combinations, the products of the probability densities, $f[x_1(k), \cdots, x_d(k)], k = 1, \cdots, l$, on the basis of the joint probability densities, $c[u_1(k), \cdots, u_d(k)], k = 1, \cdots, l$ and marginal $\{f[x_1(k)], \cdots f[x_d(k)]\}, k = 1, \cdots, l$, respectively, were calculated, and $[\tilde{u}_1, \cdots, \tilde{u}_d]$ with the maxima of the product from $[u_1(k), \cdots, u_d(k)], k = 1, \cdots, l$ was the design combination for the given CDF of precipitation of the entire region, $\tilde{u}_0$.



## 3 Study region and data

The Guangdong Province as a study case is located in South China with a land area of $158.57 \times 10^3$ km$^2$ (illustrated in Figure 1). The entire region mostly belongs to the monsoon climate zone varying from the tropic to south sub-tropic. Annual precipitation is abundant, but uneven in terms of spatial and temporal distribution. Since China's reform and opening up in the late 1970s, the water demand of the province has been increasing with the rapid development of regional socio-economy. In 2015, the total water consumption of the province accounted for 44.31 billion m$^3$, in which approximately one half was for irrigation.

According to climate, soil type, cropping system and other management measures, the entire region marked by A0 was zoned into eight sub-regions in terms of agriculture, marked by A1 - A8 in Figure 1. Areal data of sub-regions and their paddy fields were used (see Table 2). Annual precipitation data of eight sub-regions for the period of 1953 - 2013, in terms of multi-site average, were transformed from 25 hydro-meteorological stations.

The net quotas of irrigation of individual sub-regions in association with precipitation frequency resulted from the previous field experiments of irrigation in the late twentieth century. According to the distribution and statistical properties on the basis of field experiments in a research report entitled as Annual irrigation Quota in Guangdong Province (1999), the net quotas of irrigation per unit area in the paddy fields of individual sub-regions in association with the precipitation frequency are illustrated in Figure 2. They ranged from 7, 221 to 8, 520 m$^3 \cdot$hm$^{-2}$ in terms of annual average with coefficients of variation of 0.205 - 0.293, and precipitation was regarded to follow the Pearson-III distribution whose three parameters were transformed using the given mean values and coefficients of variation marked in Figure 2.

The utilization coefficient of irrigation of Guangdong province has been at a low level, which approximately accounted for 0.46 in terms of average value in 2000 according to Water Resources Comprehensive Planning of Guangdong Province. In order to respond to national tough water management measures of China, the utilization coefficient was expected no less than 0.51 by 2020 in Tough Water Management Assessing Performance of Guangdong Province. Therefore, the utilization coefficients of eight agricultural sub-regions were uniformly predefined as a fixed value of 0.51 in this paper.

## 4 Results and discussion

### 4.1 Univariate properties and distribution of precipitation

As illustrated in Figure 3, annual precipitation in the entire region and individual sub-regions remarkably changed and was typically random. Precipitation in terms of average varied mostly from 1,500 mm to 2,000 mm except for the A5 sub-region with a larger value of 2,789 mm (see Table 2). The regional maximum of precipitation accounted for 4,071 mm that occurred in 1973 in the A5 sub-region and the regional minimum of precipitation less than 800 mm occurred in 1963 in the A6 sub-region. In general, the average precipitation of the entire region accounted for 1,835 mm with the maximum of 2,421 mm and the minimum of 1,152 mm.



The general statistical characteristics of annual precipitation are illustrated in Figure 4. The upper and lower boundaries of the box were set to the values of their percentiles as one and three quarters, $Q_1$ and $Q_3$, respectively. The red solid line refers to the median. The upper and lower boundaries extended along the dash line were further set to the values as $Q_1 + 1.5(Q_1 - Q_3)$ and $Q_3 - 1.5(Q_1 - Q_3)$, respectively. All values in the range of extended boundaries were generally regarded as normal and

otherwise abnormal as marked by the red plus sign. The box plot showed that most samples of precipitation fell within the extended boundaries except for several samples from the A1 and A6 sub-regions. The statistics of goodness-of-fit test of four alternative univariate distributions were smaller than those of the significance level of 0.05 (see Table 3), which implied that the GEV, GLO, P-III and GNO distributions fitted annual precipitation of the sub-regions and the entire region. The RMSE and AIC values among the distributions slightly differed and those of the GNO distribution were the smallest for most regions.

Then, as illustrated in Figure 5, all lines of the theoretical CDF almost overlapped the points of the empirical CDF. They demonstrated that the GNO satisfactorily fitted the frequency distributions of annual precipitation for the entire region and sub-regions. As shown in Table 4, the shape parameters of the GNO in the A1 and A2 sub-regions were clear minus, that in the A5 sub-region was clear positive, and others were close to zero, which implied significantly left-skewed, right-skewed, and normal distributions, respectively.

**4.2 Eight-dimensional joint distribution of sub-regional precipitation**

A matrix of correlation coefficients between sub-regional precipitations is illustrated in Figure 6. Due to the geographical distance and direction, the coefficients differed from different pairs of sub-regions, but most of them were quite large except between A8 and other sub-regions, such as both A8-A6 and A8-A4 less than 0.1. In actuality, the correlation coefficient implied dependence between sub-regional precipitation values. The larger the coefficient was, the larger the dependence was. The Q-

Q plot of empirical and theoretical joint CDFs showed that the sample points fell near the diagonal of 1:1 even though more in the lower tail (see Figure 7). For the goodness-of-fit test of the eight-dimensional meta-Gaussian copula, the $P$-value accounted for 0.262 beyond the significance level of 0.05. That demonstrated that the high dimensional Gaussian copula better fitted the joint distribution of precipitation of eight sub-regions. However, the maximum of eight-dimensional joint CDFs was less than 0.75 using observed data.

When the conventional joint CDF of sub-regional precipitation was transformed into the Kendall CDF, the CDF was indeed enlarged (see Figure 8(a)). For example, using observed data, the minimum of 0.00005 and the maximum of 0.71957 for the conventional joint CDF corresponded to 0.00916 and 0.99084 for the Kendall CDF, respectively. Using the Hessian axes in which the scales of dual axes were transferred following the standard normal distribution (see Figure 8(b)), the conventional joint CDF and the Kendall CDF showed a linear relationship, which demonstrated that it was appropriate to use the latter

instead of the former.





### 4.3 Relationship between sub-regional joint and entire region CDFs of precipitation

According to the determined eight-dimensional Gaussian copula, a million combinations of CDFs of sub-regional precipitation were generated by using Monte Carlo simulation. Correspondingly, the conventional joint and Kendall CDFs of sub-regions and the CDFs of the entire region were achieved (illustrated in Figure 9). Comparing the conventional joint and the entire region CDFs (see Figure 9(a)), the combination points preferred to happen in the northwest and had an up-convex lower boundary. When the CDF of the entire region was given a certain value, the corresponding joint CDFs varied within the limited upper bound on which the joint CDFs were less than the given value of the entire region.

Using the Kendall CDF instead of the conventional joint CDF (see Figure 9(b)), the combination points scattered near the diagonal of 1:1 with a concave up lower boundary. In order to address the relationship of the Kendall CDF and the CDF of the entire region, a confidence interval (CI) was defined by the difference which deviated from the diagonal and transformed by the normal distribution (Serinaldi 2013; Volpi and Fiori, 2014). The red and blue envelope lines (see Figure 9(b)) refer to the two side bounds of CIs with probabilities of 0.50 and 0.95, respectively. It also showed that most observed samples fell within the envelope of CI with a probability of 0.95. In addition, between the Kendall CDF and the CDF of the entire region, there were large correlation coefficients of 0.9221 and 0.9153 for the observed and simulated samples, respectively. These showed that the Kendall CDF instead of the conventional CDF was convenient to analyze the relationship of the CDF between the entire region and the sub-regional joint CDF.

### 4.4 Design combinations of the CDF of sub-regional precipitation

The given CDFs of precipitation of the entire region can be predefined to change in the range from 0.05 to 0.95 with a step of 0.05, which refers to the alteration of regional precipitation from extreme dry to extreme wet. Considering the uncertainty of Monte Carlo simulation, those simulated combinations (see grey points in Figure 10), in which the allowable relative error of their calculated CDFs of the entire region compared to a given CDF was less than 0.05%, were selected to be an alternate for further design combination.

Using the EF method (see the blue dash in Figure 10), design points consisted of a better smooth curve which mostly fell within 0.5 of the CI. The Kendall CDF was greatly close to the CDF of the entire region, but that of the former was larger than that of the latter with lower CDFs being less than 0.55 in the study case, and vice versa in larger CDFs. Using the TY method (see the red cycles in Figure 10), design points were irregularly scattered on the two sides of the diagonal of 1:1, and even several points were out of 0.95 of the CI, for example for the given CDFs of 0.8 and 0.9. Using the MLW method (see the blue triangles in Figure 10), design points fell in the range between 0.5 and 0.95 of the CI, and design Kendal CDFs were larger than the given CDFs.

In general, if the CDF of sub-regional precipitation was equalized, the differences between the design Kendall CDF and the given CDF of the entire region were almost no more than a CI of 0.5. On the basis of maximum joint probability density, design Kendall CDFs were larger than the given CDFs, and preferred to the upper limited bound. By zooming in or out





according to the typical year, design points almost fell within 0.95 of the CI, but they were relatively scattered by comparison with other methods.

In addition, between individual CDFs of eight sub-regional precipitations and the CDF of the entire region, the design points maintained a smooth curve and were undifferentiated for all sub-regions when using the EF method (see the blue dash in Figure 11). The design individual CDFs of sub-regions were very close to the CDF of the entire region, but the former were larger than the latter in the lower CDFs and vice versa in the larger CDFs. Using TY and MLW methods, design points were scattered on the two sides of the diagonal of 1:1. Design CDFs differed from sub-regions, but their differences were undetermined. However, as seen from the envelope of design points of individual sub-regions, the ranges of the MLW design were comparatively narrow and concentrated around the diagonal of 1:1 except for A8 sub-region, but those of the TY design were much wider, in particular for the A4, A7 and A8 sub-regions.

### 4.5 Design water demand of irrigation

For given CDFs of precipitation of the entire region, the water demand of irrigation of the entire region for all selected simulations and design points are illustrated in Figure 12(a). As the given CDF changed from 0.05 to 0.95, the average value of water demand decreased from 22.79 to 12.78 billion m3, correspondingly representing from extreme dry to extreme wet (see the black dash line in Figure 12(a)). The difference between the maximum and minimum demands for a given CDF (see the blue and red dash lines in Figure 12(a)) varied in the range of 1.15 - 1.72 billion m3. Design demands of the MLW and EF methods (see the blue and black solid lines in Figure 12(a)) were slightly smaller than the average values, but those of the TY method (see the red line in Figure 12(a)) fluctuated around the line of average value. Compared to the average values (see Figure 12(b)), the maximum and minimum demands increased and decreased by 3.0% - 7.5% and 2.5% - 3.8%, respectively. Design demands of the MLW and EF methods decreased within 1.4% and 2.1%, respectively, and those of the TY method increased or decreased within 2.8% or 2.1%, respectively. These demonstrated that the differences of water demand among three design methods for the entire region were quite small.

Design water demands of irrigation in individual sub-regions are illustrated in Figure 13. As the given CDF changed from 0.05 to 0.95, sub-regional demands of the EF method smoothly decreased from 2.16-3.73 to 1.16-2.29 billion m$^3$, correspondingly representing from extreme dry to extreme wet (see the black lines in Figure 13). Then, the demands of the TY and MLW methods fluctuated around the lines of the demand of the EF method, and the fluctuations of the former were remarkably larger than those of the latter (see the red and blue lines in Figure 13). Compared to the EF method (see Figure 14), the increase or decrease of water demand of the MLW design accounted for less than 13% in most sub-regions except for A8 sub-region with the maximum of 26.1%, but that of the TY method accounted for 15.4%-24.3% for A1, A2, A3 and A5 sub-regions, and 39.8%-45.7% for A4, A6, A7 and A8 sub-regions. These demonstrated that the alterations of design water demand in sub-regions were much complicated in comparison with those in the entire region.





## 5 Conclusions

Using Guangdong province of South China as a case study of a large region, a high-dimensional meta-Gaussian copula was applied for fitting the joint distribution of multiple regional precipitation. A large number of combinations of CDFs of precipitation of eight sub-regions were generated by using the Monte Carlo method. The relationship among the CDF of the entire region, the conventional joint CDF, and Kendall CDF of sub-regions was determined. Three design methods, including the EF and TY design methods, and a new design procedure of the MLW method in association with the joint probability density, were used for design combinations of sub-regional CDFs for given CDFs of precipitation of the entire region. Then, design water demands of irrigation of the entire region and individual sub-regions were compared. The main conclusions of this study are follows:

**(1)** The frequency distributions of annual precipitation of the entire region and of sub-regions were fitted well by the GNO distribution. The shape parameters in A1 and A2 sub-regions were clear minus, those in the A5 sub-region was clear positive, and others were close to zero, which implied significantly left-skewed, right-skewed and normal distributions, respectively. The eight-dimensional Gaussian copula satisfactorily fitted the joint distribution of sub-regional precipitation.

**(2)** There was a clear linear dependence between the conventional joint and Kendall CDFs of sub-regional precipitation when both of them were transferred by the standard normal distribution. Comparing the Kendall CDFs of sub-regions and the CDF of the entire region, most observed samples fell within the envelope of CI of a probability of 0.95 around the diagonal of 1:1, and there were greater dependences between them with correlation coefficients of 0.9221 and 0.9153 for the observed and simulated samples, respectively. The use of the Kendall CDF instead of the conventional joint CDF can better link the joint frequency of sub-regions and the univariate frequency of the entire region. However, any one given CDF of the entire region corresponded to a large number of joint CDFs varying from very small to limited large. That is, there was an upper bound in larger values of the joint CDFs of sub-regions corresponding to given CDFs of the entire region.

**(3)** For given CDFs of precipitation of the entire region, design Kendall CDFs and individual CDFs of eight sub-regions of the EF method maintained a smooth curve and were very close to their diagonal of 1:1. The design Kendal CDFs of the MLW method which were larger than the given CDFs of the entire region fell between 0.5 and 0.95 probabilities for the CI far from the diagonal, and those of the TY method were irregularly scattered on the two sides of the diagonal. Then, design CDFs of individual sub-regions of the MLW and TY methods were also scattered on the two sides of the diagonal, but they differed for individual sub-regions. The change ranges of the MLW design were comparatively narrow and concentrated around the diagonal, but those of the TY design were much wider.

**(4)** For given CDFs varying from 0.05 to 0.95 representing from extreme dry to extreme wet, the simulated water demand of irrigation of the entire region in terms of the average value accounted for from 22.79 to 12.78 billion $m^3$. Design demands of the MLW and EF methods were slightly smaller than the average values and those of the TY method fluctuated around the average values. Compared to the average demand, design demands of the MLW, EF and TY methods decreased or increased,





respectively, within 1.4%, 2.1%, and 2.8%, which demonstrated that the differences of design demand of the entire region among three methods were quite small.

**(5)** For given CDFs varying from 0.05 to 0.95 representing from extreme dry to extreme wet, design water demands of individual sub-regions of the EF method decreased smoothly from 2.16-3.73 to 1.16-2.29 billion $m^3$, and those of the MLW and TY methods fluctuated around the demand lines of the EF method, but the fluctuations of the TY method were remarkably larger than those of the MLW method. Compared to the EF method, the increase or decrease of water demand of the MLW design accounted for less than 13% in most sub-regions except for the A8 sub-region with the maximum of 26.1%, but those of the TY method accounted for 39.8%-45.7% for the A4, A6, A7 and A8 sub-regions. These demonstrated that the alterations of design water demand in sub-regions were much complicated in comparison with those in the entire region.

All in all, in practical planning of regional water resources, using the EF method can realize water demand of irrigation for the entire region and its sub-regions for a given frequency of precipitation, but it is arbitrary that the series of sub-regional precipitation are regarded to be undifferentiated for a large region, for example the case region. The TY method was constrained by the limited observed data of precipitation and it cannot be chosen when several different combinations of sub-regional precipitation made the frequency of precipitation of the entire region approximately identical. Therefore, a design procedure using the MLW method in association with a high-dimensional copula, which simulated individual univariate distributions, captured their dependences for multi-variables and built a linkage between regional frequency and sub-regional frequency of precipitation, is recommended for design water demand of irrigation for a large region.

## Acknowledgements

Supported by the National Key R&D Program of China (2017YFC0405900) and the National Natural Science Foundation of China (51479217) are gratefully acknowledged.

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





**Table 1: Theoretical cumulative distribution and probability density function of three parameter univariate distribution.**

| Name | Cumulative distribution function | Probability density function | Parameter |
|---|---|---|---|
| GEV | $\exp\{-[1+\xi(x-\mu)/\sigma]^{-1/\xi}\}$ | $\frac{1}{\sigma}[1+\xi(x-\mu)/\sigma]^{-(\xi+1)/\xi}\exp\{-[1+\xi(x-\mu)/\sigma]^{-1/\xi}\}$ | $\mu \in R, \sigma > 0, \xi \in R$ |
| GLO | $1/\{1+[1-\xi(x-\mu)/\sigma]^{1/\xi}\}$ | $\frac{1}{\sigma}[1+\xi(x-\mu)/\sigma]^{-(\xi+1)/\xi}/\{1+[1+\xi(x-\mu)/\sigma]^{1/\xi}\}^2$ | $\mu \in R, \sigma > 0, \xi \in R$ |
| P-III | $\frac{\beta^\alpha}{\Gamma(\alpha)}\int_{a_0}^{x}(x-a_0)\exp[-\beta(x-a_0)]/dx$ | $\frac{\beta^\alpha}{\Gamma(\alpha)}(x-a_0)^{\alpha-1}\exp[-\beta(x-a_0)]$ | $a_0 \in R, \alpha > 0, \beta > 0$ |
| GNO | $\Phi(y), y = -\ln[1-\xi(x-\mu)/\sigma]/\xi$ | $\phi(y)/[\sigma-\xi(x-\mu)], y = -\ln[1-\xi(x-\mu)/\sigma]/\xi$ | $\mu \in R, \sigma > 0, \xi \in R$ |

**Table 2: Area, paddy field and precipitation of sub-regions and entire region. The A0 region refers to the entire region of Guangdong province.**

| Region | Area / $10^3$ km$^2$ | Paddy field/ $10^3$ hm$^2$ | Annual precipitation / $10^3$ mm | | | |
|---|---|---|---|---|---|---|
| | | | Maximum | Mean | Median | Minimum |
| A1 | 27.99 | 164.14 | 2.439 | 1.667 | 1.663 | 0.997 |
| A2 | 33.37 | 187.63 | 2.379 | 1.704 | 1.672 | 1.102 |
| A3 | 23.36 | 116.71 | 2.014 | 1.569 | 1.551 | 1.104 |
| A4 | 9.36 | 106.59 | 2.457 | 1.676 | 1.659 | 0.965 |
| A5 | 21.71 | 183.17 | 4.071 | 2.789 | 2.877 | 1.482 |
| A6 | 14.29 | 126.43 | 2.573 | 1.725 | 1.665 | 0.724 |
| A7 | 16.95 | 167.98 | 2.734 | 1.911 | 1.911 | 1.196 |
| A8 | 11.53 | 108.08 | 2.275 | 1.514 | 1.487 | 0.747 |
| A0 | 158.57 | 1160.73 | 2.421 | 1.835 | 1.845 | 1.152 |

**Table 3: Goodness-of-fit test of univariate distributions of precipitation of sub-regions and the entire region. The standard (value) refers to the statistic of the significance level of 0.05.**

| Region | GEV | | | | GLO | | | | P-III | | | | GNO | | | |
|---|---|---|---|---|---|---|---|---|---|---|---|---|---|---|---|---|
| | Statistic | Standard | RMSE | AIC | Statistic | Standard | RMSE | AIC | Statistic | Standard | RMSE | AIC | Statistic | Standard | RMSE | AIC |
| A1 | 0.065 | 0.095 | 0.028 | -426 | 0.053 | 0.099 | 0.022 | -457 | 0.065 | 0.096 | 0.027 | -430 | 0.064 | 0.096 | 0.027 | -431 |
| A2 | 0.055 | 0.095 | 0.019 | -472 | 0.075 | 0.099 | 0.026 | -434 | 0.058 | 0.095 | 0.020 | -468 | 0.058 | 0.095 | 0.020 | -467 |
| A3 | 0.051 | 0.095 | 0.019 | -473 | 0.064 | 0.098 | 0.026 | -438 | 0.049 | 0.096 | 0.020 | -468 | 0.049 | 0.094 | 0.020 | -468 |
| A4 | 0.060 | 0.094 | 0.021 | -464 | 0.057 | 0.099 | 0.022 | -455 | 0.059 | 0.096 | 0.021 | -465 | 0.059 | 0.094 | 0.021 | -465 |
| A5 | 0.087 | 0.095 | 0.030 | -419 | 0.091 | 0.098 | 0.030 | -420 | 0.087 | 0.095 | 0.028 | -426 | 0.087 | 0.095 | 0.028 | -426 |
| A6 | 0.076 | 0.094 | 0.031 | -413 | 0.090 | 0.098 | 0.032 | -410 | 0.079 | 0.095 | 0.031 | -414 | 0.079 | 0.094 | 0.031 | -415 |
| A7 | 0.063 | 0.095 | 0.025 | -440 | 0.081 | 0.097 | 0.039 | -388 | 0.068 | 0.095 | 0.028 | -426 | 0.068 | 0.095 | 0.028 | -426 |
| A8 | 0.066 | 0.095 | 0.028 | -429 | 0.088 | 0.098 | 0.040 | -384 | 0.069 | 0.095 | 0.029 | -421 | 0.070 | 0.096 | 0.030 | -420 |
| A0 | 0.076 | 0.094 | 0.022 | -454 | 0.067 | 0.098 | 0.023 | -452 | 0.072 | 0.095 | 0.022 | -458 | 0.072 | 0.095 | 0.022 | -458 |

**Table 4: Parameters of the GNO distribution of precipitation of sub-regions and the entire region.**

| Parameter | A1 | A2 | A3 | A4 | A5 | A6 | A7 | A8 | A0 |
|---|---|---|---|---|---|---|---|---|---|
| Location | 1.645 | 1.684 | 1.575 | 1.663 | 2.831 | 1.711 | 1.915 | 1.497 | 1.841 |
| Scale | 0.302 | 0.289 | 0.234 | 0.369 | 0.534 | 0.337 | 0.406 | 0.339 | 0.275 |
| Shape | -0.141 | -0.136 | 0.052 | -0.073 | 0.156 | -0.082 | 0.018 | -0.100 | 0.045 |

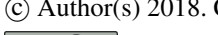



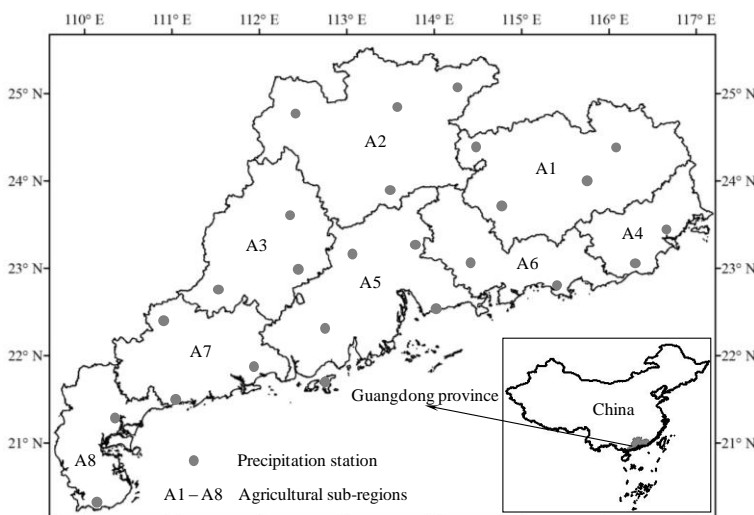

**Figure 1: Location of the study region, agricultural sub-regions and precipitation stations.**

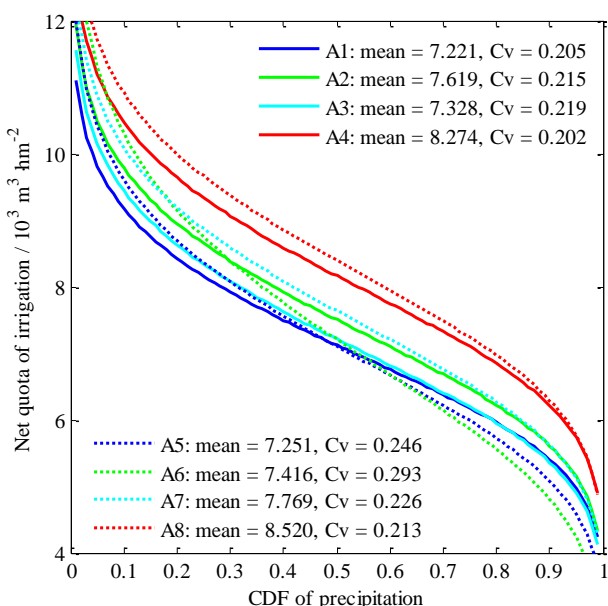

**Figure 2: Net quota of irrigation for paddy fields in individual sub-regions.**



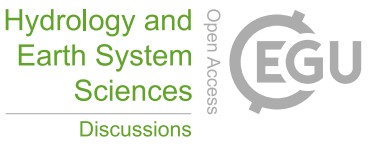

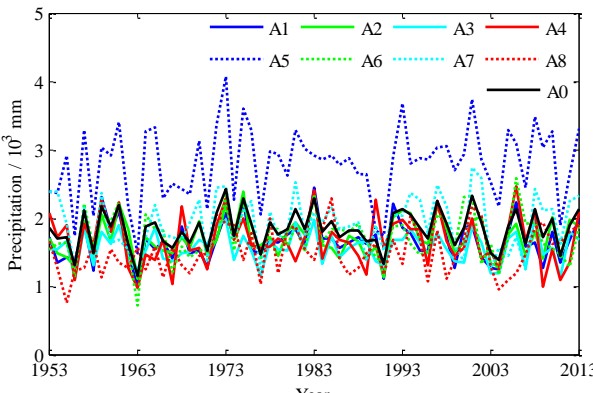

**Figure 3: Annual precipitation of sub-regions and the entire region.**

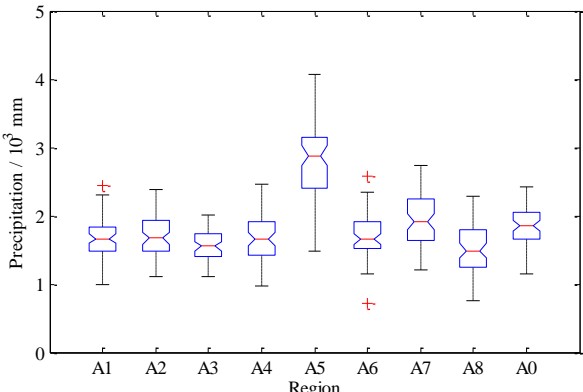

**Figure 4: Box plot of precipitation of sub-regions and the entire region.**





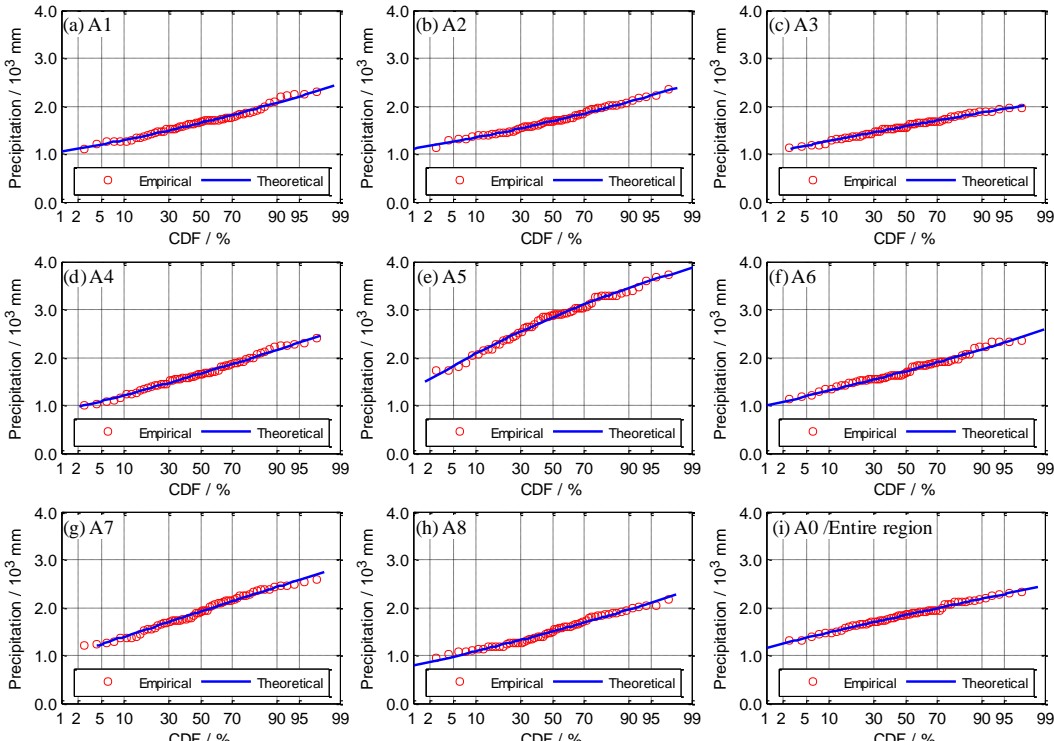

**Figure 5: Empirical and theoretical CDFs of precipitation of sub-regions and the entire region using the GNO distribution.**

|      | A1    | A2    | A3    | A4    | A5    | A6    | A7    | A8    |
|------|-------|-------|-------|-------|-------|-------|-------|-------|
| A1   | 1     | 0.839 | 0.651 | 0.745 | 0.604 | 0.765 | 0.435 | 0.148 |
| A2   | 0.839 | 1     | 0.722 | 0.611 | 0.677 | 0.599 | 0.599 | 0.238 |
| A3   | 0.651 | 0.722 | 1     | 0.572 | 0.753 | 0.610 | 0.709 | 0.309 |
| A4   | 0.745 | 0.611 | 0.572 | 1     | 0.466 | 0.646 | 0.407 | 0.032 |
| A5   | 0.604 | 0.677 | 0.753 | 0.466 | 1     | 0.708 | 0.603 | 0.264 |
| A6   | 0.765 | 0.599 | 0.610 | 0.646 | 0.708 | 1     | 0.338 | 0.079 |
| A7   | 0.435 | 0.599 | 0.709 | 0.407 | 0.603 | 0.338 | 1     | 0.489 |
| A8   | 0.148 | 0.238 | 0.309 | 0.032 | 0.264 | 0.079 | 0.489 | 1     |

5    **Figure 6: Matrix of correlative coefficients of precipitation between two sub-regions.**



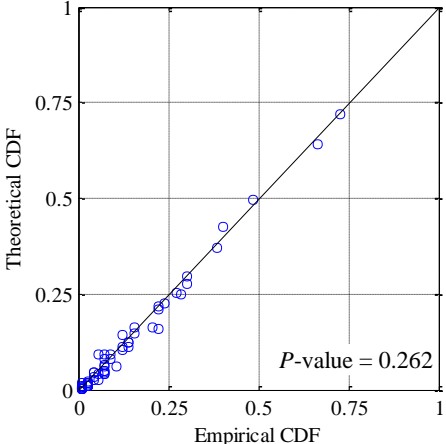

**Figure 7: Q-Q plot of empirical and theoretical CDFs of sub-regional precipitation using the eight-dimensional Gaussian copula.**

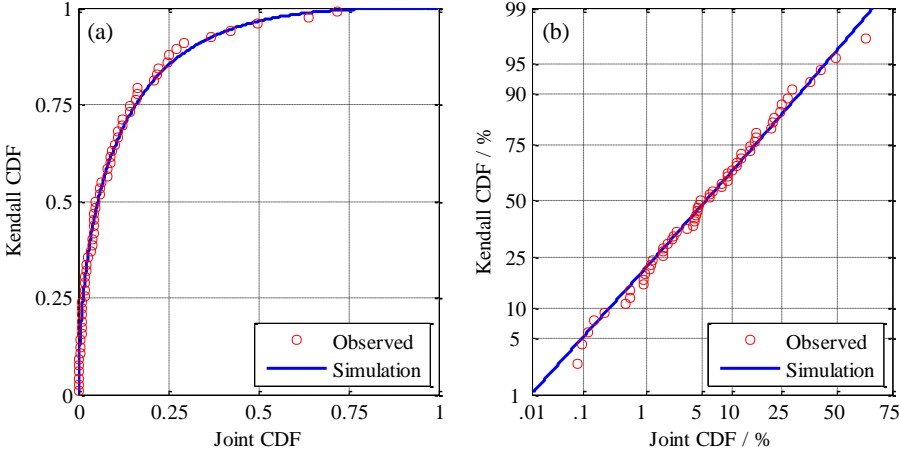

5  **Figure 8: Comparison of conventional joint CDF and Kendall CDF in subfigures (a) with the general axes and (b) with the Hessian axes.**



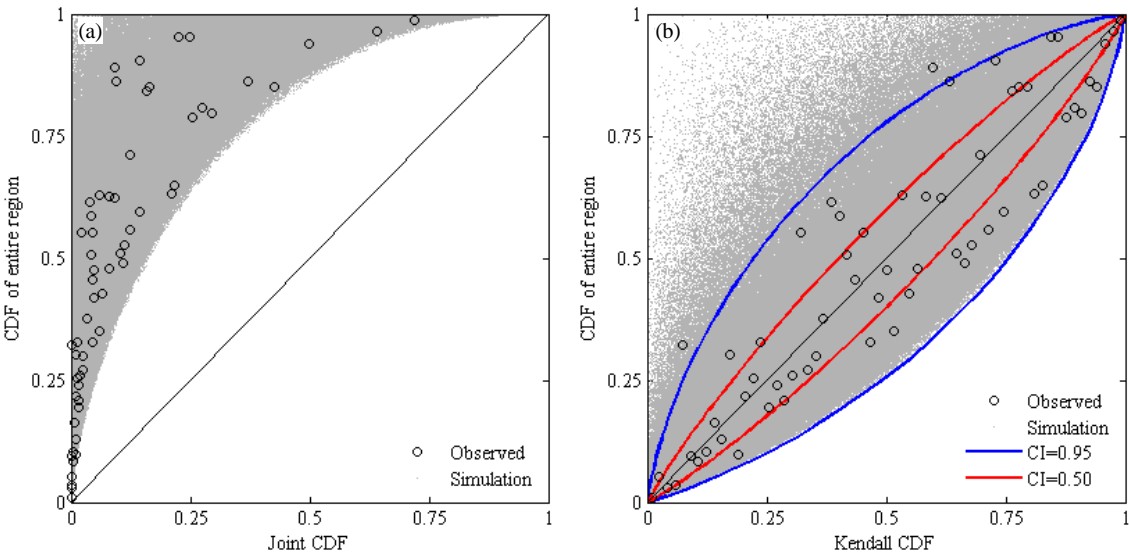

**Figure 9: CDF of precipitation of the entire region responding to (a) conventional joint CDF and (b) Kendall CDF of eight sub-regions.**

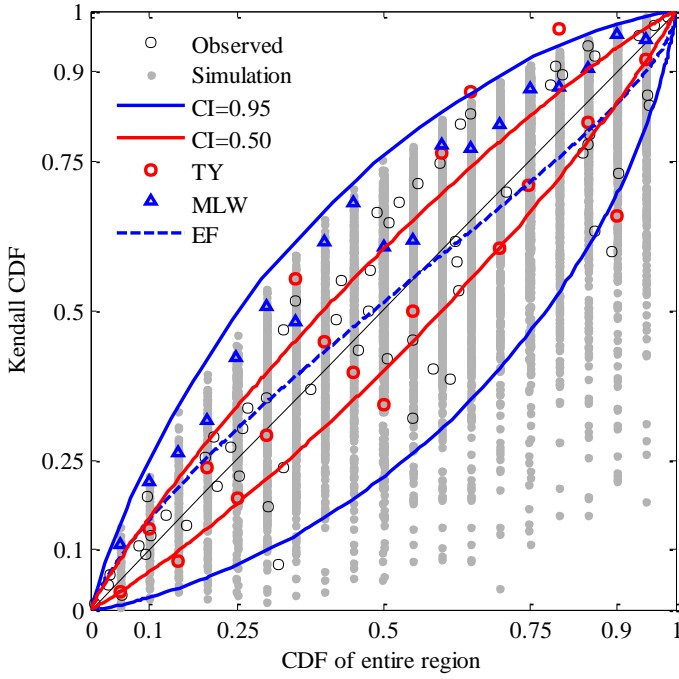

**Figure 10: Design Kendall CDFs for given CDFs of precipitation of the entire region which varied from 0.05 to 0.95 with the step of 0.05.**



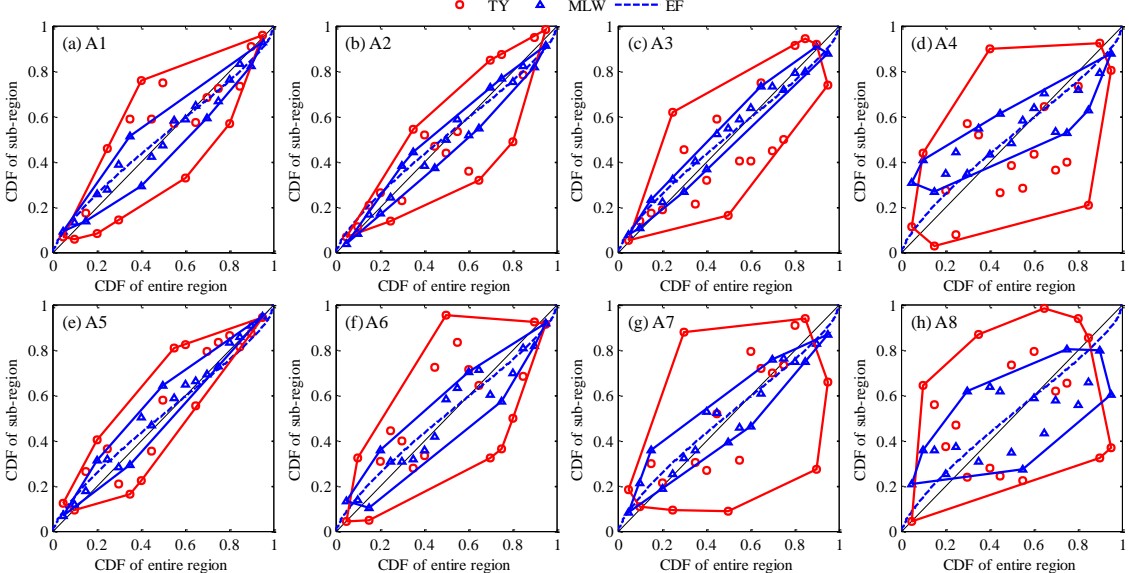

**Figure 11: Design CDFs of sub-regions for given CDFs of precipitation of the entire region.**

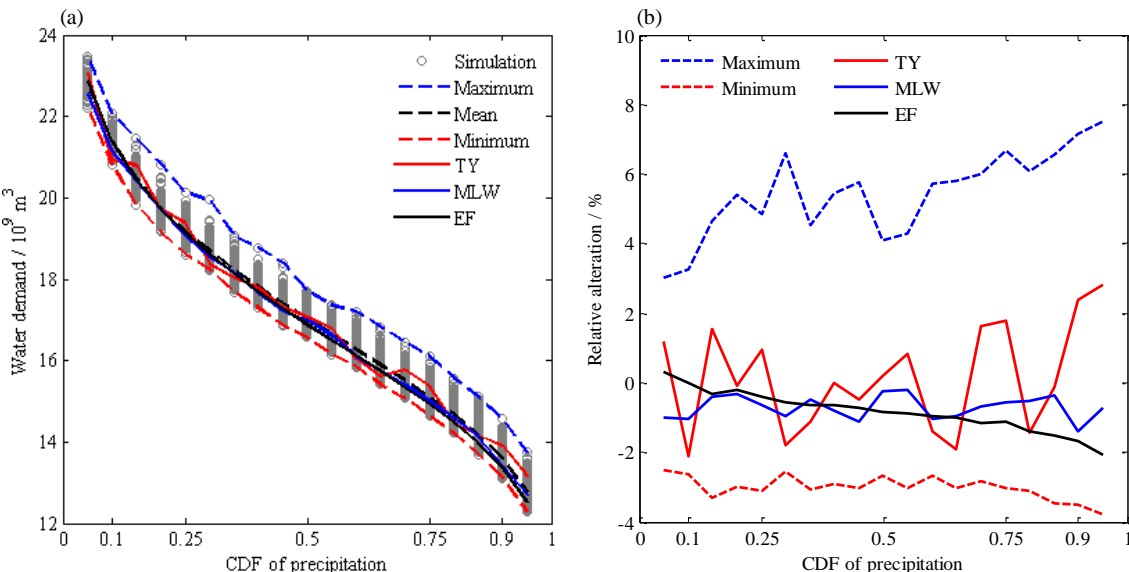

5   **Figure 12: (a) Design water demand of irrigation of the entire region and (b) their relative alteration compared to the average demand.**





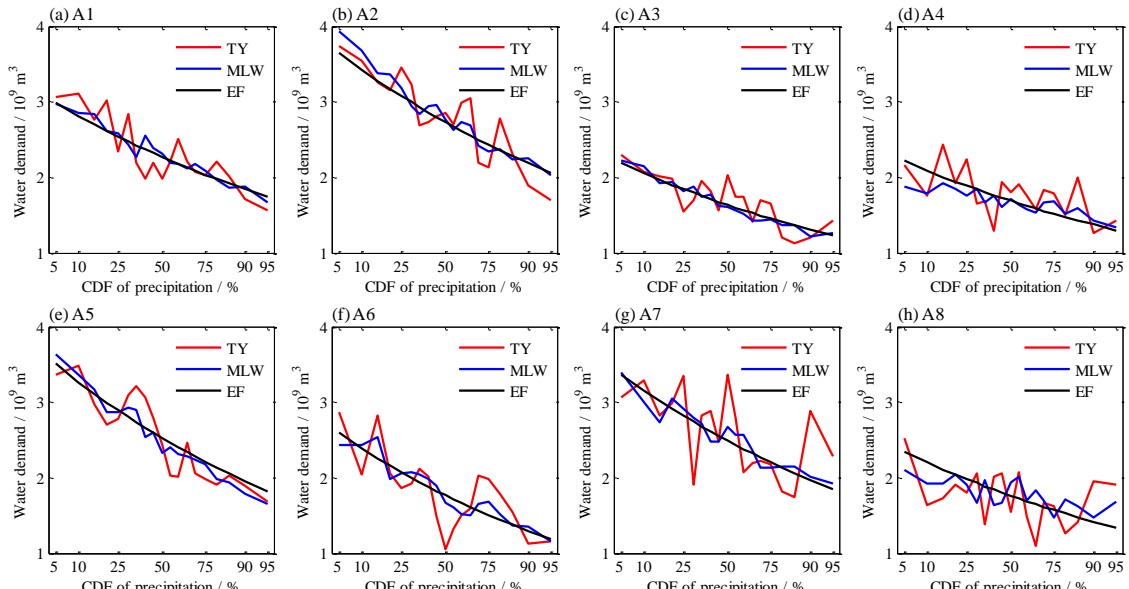

**Figure 13: Design water demand of irrigation in sub-regions for given CDFs of precipitation of the entire region.**

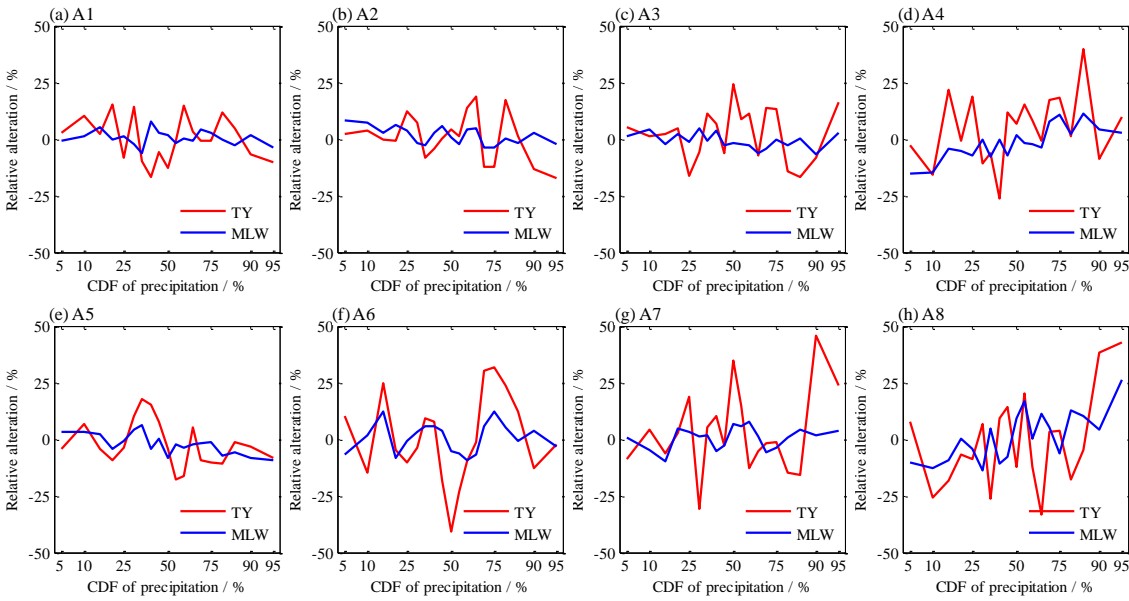

5  **Figure 14: Relative alteration of design water demand compared to the EF method for given CDFs of precipitation of the entire region.**