# Peer review of "Design water demand of irrigation for a large region using a high-dimensional Gaussian copula"

_Hydrology and Earth System Sciences, 2018_

## Referee Comment (RC1) · Anonymous Referee #1 · 7 Jun 2018

Comments on the manuscript hess-2018-213 "Design Water Demand of Irrigation for a Large Region Using a High-dimensional Gaussian Copula" by Xinjun Tu, Yiliang Du, Vijay P Singh, Xiaohong Chen, Kairong Lin, Haiou Wu

General comment The authors developed an eight-dimensional joint distribution of sub-regional precipitations using Gaussian copula, and proposed a design procedure for water demand of irrigation of a large region and provided three design methods, i.e. equalized frequency, typical year and most-likely weight function, to compare water demands of irrigation in the entire region and its sub-regions. The paper attempts to seek a new method to better design water demand of irrigation of sub-regions for a given

[Figure]

CDFs in a large region. The design procedure using the most-likely weight based on a newly-developed high-dimensional joint distribution and the linkage between regional and sub-regional frequencies of precipitation are impressive and are of novelty. The conclusions were appropriately supported by analyses results. Besides, this paper was well organized. All in all, I would like to recommend accepting this manuscript after minor revisions.

Specific comments P1.L15: The sentence "The Kendall frequency was better than the conventional joint frequency to analyze the linkage between the frequency of the entire region and the joint frequency of sub-regions." is not clear. The object of probability distribution is of precipitation or water demand of irrigation? P8.L2-4: This sentence is confusing. Please kindly explain it in detail about the using of those methods. P10. L17: The expression should be more refined. e.g., the coefficients varied from pairs of sub-regions. P10. L24: what purpose did the authors illustrate the maximum of 8-dimensional joint CDFs for? Please kindly give more details, or not, I suppose it should be considered to delete. P10. L29: the pronouns (the latter and the former) are a little bit ambiguous. They represented conventional joint CDF and the Kendall CDF, relatively, or dual axes and Hessian axes relatively? Apart from that, please kindly explain what aspects were the latter more suitable than the former? P20: The contents illustrated in Figures 3 and 4 are similar. Kindly recommend deleting one of them.

---

## Referee Comment (RC2) · Anonymous Referee #2 · 3 Jul 2018

General comment In this paper, authors used the multivariate Gaussian copula and the general normal distribution to develop an eight-dimensional joint distribution of sub-regional precipitations. Using three design methods, i.e. equalized frequency, typical year and most-likely weight function, design combinations of sub-regional precipitation for a given cumulative frequency of entire regional precipitation were proposed and applied to analyze water demand of irrigation in a large region and its sub-regions. In a large region, design combinations of sub-regional water demand of irrigation were produced by the linkage between the regional CDF of precipitation and the joint CDF of sub-regional precipitation, which is impressive and innovative. The technical methods are overall sound, and the recommended design approach is useful for the water resource managers in long-term planning. I would recommend accepting this manuscript after the following concerns have been fully addressed.

Specific comments 1. [P6: 2.4.1], the Gaussian copula may not work for all cases, did authors consider other available copula functions at higher dimensions? For instance, t-copula is conceptually similar to Gaussian copula and also available at higher dimensions. 2. [P9: Lines 8-11], why is the entire region divided into eight sub-regions? Are there any references that authors can provide to support agricultural division in the study region? 3. [P10: Lines 1-5], the explanation of the box plot should be addressed in the title of Figure 4. 4. [P11: Lines 10-14], the confidence interval (CI) was mentioned many times after here, why use it and how to calculate it, should be addressed in Methodology. 5. [P20: Figures 3 and 4], the implication of Figures 3 and 4 is similar. Figure 3 may be deleted. 6. [P21: Figure 5], the label of X-axis is unconventional. Kindly suggest using general axis for the X-axis or giving an explanation about it. 7. [P22: Figure 7], how to calculate multivariate empirical CDF? 8. [P22: Figure 8], the illustrations of two subfigures are undifferentiated except the ticks of X-axis. Delete one of them.

---

## Author Comment (AC1) · 4 Jul 2018

Anonymous Referee #1 Comments on the manuscript hess-2018-213 "Design Water Demand of Irrigation fora Large Region Using a High-dimensional Gaussian Copula" by Xinjun Tu, Yiliang Du,Vijay P Singh, Xiaohong Chen, Kairong Lin, Haiou Wu General comment The authors developed an eight-dimensional joint distribution of sub-regional precipitations using Gaussian copula, and proposed a design procedure for water demand of irrigation of a large region and provided three design methods, i.e. equalized frequency, typical year and most-likely weight function, to compare water demands of irrigation in the entire

region and its sub-regions. The paper attempts to seek a new method to better design water demand of irrigation of sub-regions for a given CDFs in a large region. The design procedure using the most-likely weight based on a newly-developed high-dimensional joint distribution and the linkage between regional and sub-regional frequencies of precipitation are impressive and are of novelty. The conclusions were appropriately supported by analyses results. Besides, this paper was well organized. All in all, I would like to recommend accepting this manuscript after minor revisions. Specific comments P1.L15: The sentence "The Kendall frequency was better than the conventional joint frequency to analyze the linkage between the frequency of the entire region and the joint frequency of sub-regions." is not clear. The object of probability distribution is of precipitation or water demand of irrigation? Response: We greatly thank the reviewer for the comment and revised the statement. The probability distribution refers to that of precipitation. (see Page 1, Lines 15-16 in the revised manuscript). P8.L2-4: This sentence is confusing. Please kindly explain it in detail about the using of those methods. Response: We greatly thank the reviewer for the comment and revised the sentence (see Page 8, Lines 5-6 in the revised manuscript). P10.L17: The expression should be more refined. e.g., the coefficients varied from pairs of sub-regions. Response: We greatly thank the reviewer for the comment and revised it (see Page 10, Line 20 in the revised manuscript). P10. L24: what purpose did the authors illustrate the maximum of 8-dimensional joint CDFs for? Please kindly give more details, or not, I suppose it should be considered to delete. Response: We greatly thank the reviewer for the comment and revised the sentence. We would like to present the maximum in order to point out the limit of the conventional joint CDFs. Using the Kendall frequency can break through the limit (see Page 10, Lines 27-32 and Page 11, Lines 1-2 in the revised manuscript). P10. L29: the pronouns (the latter and the former) area little bit ambiguous. They represented conventional joint CDF and the Kendall CDF, relatively, or dual axes and Hessian axes relatively? Apart from that, please kindly explain what aspects were the latter more suitable than the former? Response: We greatly thank the reviewer for the comment and revised the statement

(see Page 11, Lines 1-2 in the revised manuscript). P20: The contents illustrated in Figures 3 and 4 are similar. Kindly recommend deleting one of them. Response: We greatly thank the reviewer for the comment and deleted Figure 3. Figures 4-14 in the original manuscript were revised to Figures 3-13 in the revised manuscript, respectively.

Please also note the supplement to this comment:
https://www.hydrol-earth-syst-sci-discuss.net/hess-2018-213/hess-2018-213-AC1-supplement.pdf

---

## Author Comment (AC2) · 4 Jul 2018

Anonymous Referee #2 General comment In this paper, authors used the multivariate Gaussian copula and the general normal distribution to develop an eight-dimensional joint distribution of sub-regional precipitations. Using three design methods, i.e. equalized frequency, typical year and most-likely weight function, design combinations of sub-regional precipitation for a given cumulative frequency of entire regional precipitation were proposed and applied to analyze water demand of irrigation in a large region and its sub-regions. In a large region, design combinations of sub-regional water demand of irrigation were produced by the

linkage between the regional CDF of precipitation and the joint CDF of sub-regional precipitation, which is impressive and innovative. The technical methods are overall sound, and the recommended design approach is useful for the water re-source managers in long-term planning. I would recommend accepting this manuscript after the following concerns have been fully addressed. Specific comments 1. [P6: 2.4.1], the Gaussian copula may not work for all cases, did authors consider other available copula functions at higher dimensions? For instance, t-copula is conceptually similar to Gaussian copula and also available at higher dimensions. Response: We greatly thank the reviewer for the comment. In our previous investigation, the Gaussian-copula and t-copula functions were also used. the goodness-of-fit test of joint distribution showed that the RMSE and AIC values of two copulas were almost undifferentiated (see the table as follows). Goodness-of-fit test of joint distribution of sub-regional precipitation Type P-values RMSE AIC Gaussian copula 0.262 0.0173 -401.36 t-copula 0.250 0.0173 -401.37 The manuscript paid more concerns to design procedures for water demand of irrigation. Considering the limited length of the paper, only the Gaussian copula was addressed and further applied in modelling the joint distribution of sub-regional precipitation in our manuscript. 2. [P9: Lines 8-11], why is the entire region divided into eight sub-regions? Are there any references that authors can provide to support agricultural division in the study region? Response: We greatly thank the reviewer for the comments and revised the statement (see Page 9, Lines 14-16 in the revised manuscript). Eight sub-region in terms of agriculture is based on the report of Irrigation Quota of Guangdong Province. 3. [P10: Lines 1-5], the explanation of the box plot should be addressed in the title of Figure 4. Response: We greatly thank the reviewer for the comment. The description of the box plot was moved in the title of Figure 3 in the revised manuscript. (see Page 10, Lines 8-9 and Page 20, Lines 2-5 in the revised manuscript) 4. [P11: Lines 10-14], the confidence interval (CI) was mentioned many times after here, why use it and how to calculate it, should be addressed in Methodology. Response: We greatly thank the reviewer for the comment. A confidence interval (CI) was defined by the distance which deviated

from the diagonal (Serinaldi 2013; Volpi and Fiori, 2014) and transformed herein by the normal distribution. In design sub-regional CDFs of precipitation, The CI may not be necessarily, but it is used to further illustrate the relationship of the Kendall (joint) CDF of sub-regional precipitation and the CDF of precipitation of the entire region for the samples and design values (see Figures 8 and 9 in the revised manuscript) Considering the manuscript paid more concerns to design procedures for water demand of irrigation, we gave a simple but clear definition of CI in Methodology (see Page 9, Lines 3-6 in the revised manuscript). More details of the CI can be referred in previous studies, for example by Serinaldi (2013),Volpi and Fiori (2014), etc.. 5. [P20: Figures 3 and 4], the implication of Figures 3 and 4 is similar. Figure 3 may be deleted. Response: We greatly thank the reviewer for the comment and deleted Figure 3. Figures 4-14 in the original manuscript were revised to Figures 3-13 in the revised manuscript, respectively. 6. [P21: Figure 5], the label of X-axis is unconventional. Kindly suggest using general axis for the X-axis or giving an explanation about it. Response: We greatly thank the reviewer for the comment and gave a description of the X-axis (see Page 20, Lines 8-9 in the revised manuscript). 7. [P22: Figure 7], how to calculate multivariate empirical CDF? Response: We greatly thank the reviewer for the comment. The description of multivariate empirical CDF was addressed in Methodology (see Page 7, Lines 17-18 in the revised manuscript). 8. [P22: Figure 8], the illustrations of two subfigures are undifferentiated except the ticks of X-axis. Delete one of them. Response: We greatly thank the reviewer for the comment and would maintain two subfigures. The left figures generally used in most studies clearly presented the relationship and change between the Kendall CDF and conventional joint CDF. The right figure illustrated in this paper demonstrated that both of them transformed by the standard normal distribution showed a good linear relationship. (see Page 10, Lines 30-32 and Page 11, Lines 1-2 in the revised manuscript).

Please also note the supplement to this comment:
https://www.hydrol-earth-syst-sci-discuss.net/hess-2018-213/hess-2018-213-AC2-

supplement.pdf

**Supplement:**

[revised manuscript text omitted]